# Clinical Dental Midline Shift Is Not a Predictor of the Side of Shorter Hemimandible: A Cone Beam Computed Tomography Diagnostic Study

**DOI:** 10.3390/diagnostics15020161

**Published:** 2025-01-13

**Authors:** Lígia Pereira da Silva, Alicia López-Solache, Urbano Santana-Penín, José López-Cedrún, María Jesus Mora, Pablo Varela-Centelles, Antonio González-Mosquera, Almudena Rodríguez-Fernández, Urbano Santana Mora

**Affiliations:** 1FP-I3ID, Faculty of Health Sciences, University Fernando Pessoa, 4249-004 Porto, Portugal; ligia@ufp.edu.pt; 2Department of Surgery and Medical-Surgery Specialties, University of Santiago de Compostela, 15782 Santiago de Compostela, Spain; alicia.lopez.solache@rai.usc.es (A.L.-S.); mariajesus.mora@usc.es (M.J.M.); pabloignacio.varela@usc.es (P.V.-C.); clinicaglezmosquera@gmail.com (A.G.-M.); urbano.santana.mora@usc.es (U.S.M.); 3Oral and Maxillofacial Surgery Service, University Hospital Complex of La Coruña, 15151 La Coruña, Spain; jose.lopez.cedrun@sergas.es; 4Department of Public Health, University of Santiago de Compostela, 15782 Santiago de Compostela, Spain; almudena.rodriguez@usc.es; 5Consortium of Biomedical Research in Epidemiology and Public Health (CIBER Epidemiology and Public Health—CIBERESP), 28029 Madrid, Spain; 6Health Research Institute of Santiago de Compostela (IDIS), 15782 Santiago de Compostela, Spain

**Keywords:** dental midline shift, dental occlusion, asymmetry, jaw asymmetry, jaw growth, temporomandibular disorders (TMDs)

## Abstract

**Background/Objectives**: Interincisive midline deviation is frequent. Determining the cause (dental versus skeletal) is crucial for treatment planning. This study assessed the null hypothesis that neither clinical dental midline shift nor the temporomandibular disorder (TMD)-affected side correlate with maxillary/mandibular asymmetry. **Methods**: Thirty-eight CBCT scans were analyzed: thirty-five (92.1%) females, three (7.9%) males; mean (SD) age 34.6 (11.9) years old. Tomographic images were acquired using the i-CAT^®^ Imaging System; mandibular/maxillary measurements were obtained with the Planmeca Romexis^®^ software v.6.This is an ancillary study of a clinical trial (NCT02144233) that included chronic pain (TMD diagnosis; DC/TMD criteria), fully dentate, and stable normo-occlusion participants. **Results**: We found sixteen (42.1%) dental midline deviations to the right and thirteen (34.2%) to the left. In the study population, the right side was more developed: a hemimandible length of 119.4 (5.7) mm versus 118.6 (5.3) mm for the right and left sides, respectively (95% CI 0.21 to 1.51), *p* = 0.01. **Conclusions**: Neither the dental midline shift side nor the affected side predicted a less developed hemimandible.

## 1. Introduction

Dental midline shift (DMS) is a common finding in dentistry [1]. From the perspective of dental occlusion, DMS requires an assessment beyond the central incisors [2]. It may reflect a more distal position on the side towards which the lower dental midline deviates than on the opposite side [3]; this may imply an alteration in the nature of certain contacts, which could modify the biodynamics and/or function of the stomatognathic system (SS) [4,5]. Centric occlusion should be explored to determine whether the deviation is organic or deflective in nature, corresponding to a functional deviation during jaw closure [6]. Facial asymmetry may have different origins, namely: (i) congenital (originating prenatally), (ii) developmental (originating during growth), and (iii) acquired (resulting from functional jaw displacements, traumatic injury, or other factors) [7]. However, dental asymmetries may occur with perfect facial symmetry. According to Bishara et al. (1994) [8], facial asymmetry can be classified as dental, skeletal, muscular, functional, or combinations thereof. Once the deviation is clinically evident (Figure 1), the question is whether it is the upper arch (i.e., maxillary bone) that deviates from the mid-sagittal plane, whether the deviation is due to the mandibular structures, or both. Dental midline asymmetries demand specific consideration in rehabilitation (orthodontic or prosthetic procedures) during diagnosis and treatment planning, since they represent one of the most challenging, frequent, and enduring issues that dentists deal with [9,10].

Asymmetry of the mandible is a condition that exists when the right and left sides of the mandible are not mirror images of each other [5]. Lower facial asymmetries are a marker for environmental stress and cerebral lateralization during early development [11], and they may be the result of asymmetry of bones [12], teeth, or soft tissues [12,13]. Factors such as tooth loosening, sleep position, or altered chewing function can contribute to facial asymmetry [14,15]. In particular, it has been shown that the volume of the chin becomes larger on the side opposite the habitual chewing side [15].

Knowledge of the possible cause is important, and to reach the correct diagnosis, the dentofacial structures must be properly examined. If the cause is solely dental, treatment can be approached from an occlusal perspective. However, if the cause is basal bone asymmetry, treatment may vary [16,17]. Mandibular development takes place during growth, but it is virtually non-existent during adulthood [18]. Accordingly, the correction or prevention of bony asymmetry could take place during development, but not during adulthood [18,19]. It seems logical that it would be of interest to determine whether the shift in the dental midline is exclusively dental in origin (involving only the alveolar bone) or whether it is a consequence of asymmetry of the basal bones of the SS.

The most accurate method that can be used to determine the size of the bony structures is cone beam computed tomography (CBCT) [20]. This procedure is not absolutely risk-free, meaning it should not be performed on healthy individuals [21]. The indication for CBCT for other reasons, such as certain types of TMDs, allows for measurements of the SS, specifically to assess the dental midline shift.

Interincisive midline is a clinical sign that carries implications in various diagnoses (orthodontic and rehabilitative, as well as in pediatric dentistry). In this study, we considered clinical dental asymmetries as the sole parameter and then evaluated whether there was also a skeletal problem. We also speculated that the hemimandible on the affected side was shorter than that on the opposite (unaffected) side. In the case of TMDs, the previous literature has reported a tendency to preferential chewing on the affected side [22,23,24]. This phenomenon could lead to a greater development of the non-chewing side bone size. Therefore, it is logical to assume that mandibular asymmetry could be a factor associated with the TMD-affected side. Previous studies on mandibular asymmetry focused mainly on the asymmetry of temporomandibular joints, and not on mandibular growth [25].

We hypothesize that when dental midline shift is present, the hemimandible on the side towards which the mandibular dental midline deviates is shorter than the hemimandible on the opposite side.

## 2. Materials and Methods

### 2.1. Ethics Statement

This study was approved by the Regional Ethics Committee (CAEI, approval number 2009/017; updated on 29 November 2013). Informed consent was obtained from all participants, and the data released here are consistent with the consent obtained.

This study is an analysis of diagnostic aspects of TMDs that were pre-specified in the MAP Protocol: Restoring physiological jaw closure and masticatory function as treatment for chronic facial pain: A randomized clinical trial (ClinicalTrials.gov NCT02144233).

### 2.2. Participants

Participants in the MAP trial were eligible for this study. A prior diagnosis of chronic TMD, according to DC/TMD criteria [26], was required. The tomographic images were obtained using the 17–19 i-CAT^→^ Imaging System (Imaging Sciences International Inc., Hatfield, PA, USA). The criterion for CBCT exploration in the MAP study was determined by chronic TMD diagnostic needs. The following criteria were established for the selection and collection of data.

#### 2.2.1. Inclusion Criteria

Persons with chronic TMDs refractory to conservative treatment, referred to a tertiary health care center, aged between 18 and 65 years. Participants were required to have experienced chronic pain (minimum 6 months’ duration) diagnosed as TMD, with a self-reported intensity of moderate to severe (≥4 to ≤9) scores on a visual analog scale graded from 0 to 10, where 0 = no pain and 10 = worst possible pain [27,28].

The diagnosis of TMD was established according to the DC/TMD criteria [26]. Pain was presumed to be relevant since the patient demanded treatment of any kind. An examination of the dental status and occlusion of the participants was also performed. To avoid confounding factors, only fully dentate subjects (but third molars) or participants with fixed partial dentures allowing comfortable chewing with clinically normal or near-normal occlusion (stable Angle class I) were selected. Dental midline deviation was qualitatively assessed by two experienced trialists (disagreements solved by discussion) using frontal digital telephotographs where mandibular interincisive midline shift (left or right) was registered using the maxillary dental midline as a reference. All participants in the study were carefully evaluated both clinically and through computerized kinesiography (K7, Myotronics Inc., Kent, WA, USA) from a biodynamic perspective. The displacement between centric occlusion and maximum intercuspation occurred at the expense of mandibular advancement, but with minimal or no displacement in the latero-medial direction. Therefore, the asymmetry of the lower dental midline relative to the upper midline was organic in this study population [8]. Condylar paths were assessed by conventional axiography.

#### 2.2.2. Exclusion Criteria

These criteria included oncological pathology, traumatic or neurological injuries, and congenital deformity. Participants with severe psychological disturbances (psychosis, major depression), cognitive impairment, dental health professionals, substance abuse, in litigation or claiming disability/retirement compensation for chronic pain, previous orthodontic treatment, or severe periodontal disease (grade 3 mobility) were excluded. Patients with systemic diseases that could affect the masticatory system were also excluded.

### 2.3. Data Collection/CBCT Acquisition

CBCT scans of the maxillofacial region were obtained using the 17–19 i-CAT^→^ Imaging System (Imaging Sciences International Inc., Hatfield, PA, USA) under the following parameters: 120 kVp, 0.4 mm^3^ voxel size, 8.9 s scan time, and 16.5 cm × 13.5 cm field of view. This scanning device allows patients to sit upright during the procedure, and their head position, such as the Frankfort horizontal (FH) plane, was parallel to the floor. Throughout the scanning procedure, patients were instructed to maintain light contact of their teeth with the bite-peg, and the facial soft tissues were at rest. CBCT scans were stored in the Digital Imaging and Communications in Medicine (.dcm) format.

### 2.4. Morphometric Analysis

It is important to note that conventional CBCT software has some limitations. It is not possible to directly measure certain variables such as lengths in oblique directions, e.g., the length of the hemimandible. Therefore, an image-processing software that would allow this measurement was employed. CBCT images were loaded into Planmeca Romexis^→^ dental software v.6 (Planmeca, Helsinki, Finland); the reference points and anatomical planes detailed below were identified and measured. Each measurement was independently performed by two specifically trained examiners, at different times. Firstly, all CBCT images were oriented according to two planes: the Frankfort horizontal plane (FP) and the mid-sagittal plane (MSP). This phase allowed the standardization of the method and was undertaken before any measurement.

The evaluation was completed independently and randomly in a well-lit space. The landmarks highlighted in Table 1 and Figure 2 were used, and the following measurements were performed on each tomographic image scan [29,30,31,32,33,34,35]:

Condylar process height: From point Cs to a horizontal line (parallel to FP) drawn at point SN. Measurement made on the right and left sides;Coronoid process height: From point Cp to a horizontal line (parallel to FP) drawn at point SN. Measurement taken on the right and left sides;Articular eminence (AE) height: From point GF to a horizontal line (parallel to FP) drawn at point AE. Measurement taken on the right and left sides;AE inclination: Angle formed between a line parallel to FP, drawn at point AE, with a line drawn between points AE and GF [36]. Measurement made on the right and left sides;Mandibular ramus height: From point Cs to MB1. Measurement made on the right and left sides;Mandibular body length: From point Rp to Pg. Measurement taken on the right and left sides;Hemimandible length: From point Cpt to point Pg. Measurement taken on the right and left sides (Figure 3);Right hemimaxilla width: From point P16 to MSP. Measurement made only on the right side by drawing a horizontal line (parallel to the FP) between the two references;Left hemimaxilla width: From point P26 to MSP. Measurement made only on the left side by drawing a horizontal line (parallel to the FP) between the two references;Distance Cl-MSP: From point Cl to MSP. Measurement made by drawing a horizontal line (parallel to the FP) between the two references and taken on the right and left sides.

### 2.5. Statistical Analysis

Continuous variables were described through the mean and standard deviation (SD). Dichotomic variables were described through the frequency and percentage. The normality of the distribution was assessed with Shapiro–Wilk tests, while the homogeneity of variances was verified by Levene’s test. Normally distributed continuous variables were compared using a two-tailed paired Student’s *t*-test. Similarly, if the variables did not follow a normal distribution, paired non-parametric tests were applied. A one-sample proportion test was carried out to assess possible differences in group proportions. Possible relations between variables were assessed through Pearson’s chi-square test. The reproducibility of the measurements was assessed using the intraclass correlation test (ICC). Statistical analysis was executed using SPSS (Statistical Package for the Social Sciences, IBM, v.29.0.2) [37]. The alpha level was set at *p* = 0.05.

## 3. Results

Forty-one CBCT scans were evaluated. Of this sample, it was only possible to perform the intended measurements in 38 as the CBCT images obtained in 3 of the cases had artifacts that prevented the identification of key reference points and therefore did not allow the standardization of the measurement method. Two calibrated researchers independently performed all measurements, yielding an ICC that ranged from 0.87 to 0.95. Raw data can be seen in the Appendix A.

The sample was chosen by convenience and consisted of 3 (7.9%) males and 35 (92.1%) females. Mean age 34.6 (11.9) years. Regarding the location of the symptoms, 26/38 (68.4%) persons indicated that it was unilateral, and 12/38 (31.6%) showed symptoms on both sides. The assessment of the dental midline deviation revealed that 42.1% (*n* = 16) had a deviation to the right, 34.2% (*n* = 13) to the left, and 21.1% (*n* = 8) had no clinical dental midline shift. One person (2.6%) could not be assessed (participant wearing an upper fixed partial denture, which did not allow the assessment of a dental midline between natural incisors). Table 2 shows the main biological characteristics of the subjects.

Overall (*n* = 38), statistically significant asymmetry was observed in the study group. The magnitudes of these differences appear relatively small, so that they do not a priori show great clinical significance. One of the main study outcomes was the hemimandible length. The hemimandible was shorter on the left side (intraindividual paired differences 0.9 (95% CI 0.2 to 1.5), *p* = 0.01). Other outcomes relative to the mandible also indicated a smaller size on the left side. This was not the case for the maxillary dimensions, which showed no evidence of differences between the right and left sides. Axiographic recordings showed that mean condylar path angles were higher on the lef side than on the right one (mean difference −3.1 (6.3 to 0.0), *p* = 0.05) (Table 3).

The side toward the lower dental midline shift was deviated (16 persons to the right and 13 to the left) did not predict a shorter hemimandible or asymmetry of other bone structures. However, coronoid apophysis was larger on the side opposite to the side where the dental midline shift was deviated (Table 4).

Table 5 shows mean values of bone sizes on the affected versus unaffected side. There was no evidence of differences between the affected and unaffected sides in the hemimandible length (27.07 (1.36) mm on the affected side and 26.96 (1.47) mm on the nonaffected side; mean difference 0.11; 95% CI −0.14 to 0.36; *p* = 0.36). This study revealed a tendency towards a greater mandibular ramus height on the unaffected side (*p* = 0.06). Also, the condylar path angle was higher on the affected side in this group with unilateral symptoms (*n* = 25).

## 4. Discussion

The mandible was asymmetrical in this study population, with the left hemimandible being shorter than the right. However, unexpectedly, the side toward the jaw dental midline was deviated with respect to the upper midline, and this did not predict a shorter hemimandible. Most asymmetries occur in the lower face [38]; therefore, the dental midline of the mandible was selected as a reference point rather than the facial midline to explore whether different interarch incisal asymmetries can be observed in a clinical context, although not primarily related to facial aesthetics.

The fact that the side with interincisive mandible midline deviation does not coincide with the less developed side of the jaw seems to suggest that this purely dental relationship is due to the biomechanics of the SS. These biomechanics could explain an adaptive dental repositioning to interarch occlusal forces, which could not only correspond to the developmental period but also to later periods. Since humans are sensitive to disruptions in a bilaterally symmetrical anatomy, like the face, symmetry plays a significant part in how we perceive beauty [39] and also in how jaw biodynamics and functional dental guidance occur [4]. There are currently a number of studies that have determined thresholds for dental midline shift perception [40]. For instance, a movement in the dental midline of at least 2 mm was observed by laypeople [41,42,43]. A different study found that a dental midline deviation of 2.9 (1.1) mm was no longer regarded as acceptable [39]. More recently, more precise discrepancy localization has been achieved using more advanced imaging techniques such as CBCT [20] and three-dimensional photography [15]. To ascertain the origin, degree, and location of asymmetry, a thorough examination of the patient’s numerous diagnostic records is required [1]. With their demonstrated accuracy and precision, more recent developments in three-dimensional photography have become a useful tool for the diagnosis and treatment planning of dentofacial asymmetries [44,45]. Decisions on dentofacial deformity intervention, however, are based on the patient’s awareness of the aesthetic issue, the severity of the occlusal deformity, and any associated vertical or sagittal jaw imbalance [46].

There were no differences between the size of the structures on both sides when the side with dental midline shift was considered. Hence, dental midline shift does not predict bony asymmetry. The jaw dental midline sometimes deviates towards the longer hemimandible, which presumably occurs as a function of biodynamic and masticatory function during the adult period. We therefore speculate that this type of alteration would be easier to correct, as it is only dependent on the alveolar process, which is more sensitive to orthodontic management.

There was no evidence of a relationship between the TMD-affected side and the asymmetry of hard tissues (Table 5). A tendency was observed for a larger coronoid apophysis on the side opposite the side where the jaw dental midline shift was deviated (Table 4). Coronoid apophysis provides insertion to the temporalis muscle, a positioner of the jaw. This finding seems to suggest the influence of asymmetrical muscle coactivation of the temporalis muscle. There is a tendency to use the left side more frequently in this study population. The masseter muscle on the chewing side is mainly activated; this activity requires torque-like coactivation of the contralateral musculature, mainly the temporalis muscle, to position and stabilize the jaw, which may explain the greater development of the coronoids.

Flanze et al. [40], in 2023, asserted that dental midline shifts should be addressed as soon as feasible to prevent them from impairing face growth. The authors stated that these shifts are frequently functional in nature because of a lateral forced bite of the mandible and the altered chewing function [14]. This is in agreement with Heikkinen et al. [15], who conducted a study in twins and observed that chin volume is increased on the non-habitual chewing side. These authors concluded that functional factors are more significant than genetic factors.

One of the strengths of our study is that the clinical condition of the participants was extremely homogeneous: complete natural dentition, normal or near-normal occlusion. Furthermore, the method used to perform the measurements had already been used in previous research [29,30,31,32,33,34,35] and was therefore proven to be reliable and standardized. Independent observer measurements in this study showed an ICC that ranged from 0.87 to 0.95, confirming the high reliability of the measurement method.

This study has limitations. It has a low number of male subjects, which is common in people with TMDs referred to tertiary care services [47]. The generalizability of the findings could be limited to females due to the low number of males. However, this asymmetrical female-to-male ratio is characteristic of samples from patients with TMDs referred to tertiary care centers [47]. Comparisons of anatomical structure sizes must often account for sex differences. However, in the study of asymmetry (intra-individual comparisons), no a priori differences between sexes are expected. The main purpose of this study was to answer the clinical question: Is the clinical perception that the mandible is smaller on the side to which the mandibular midline deviates accurate? A post hoc test, including only females (*n*= 35), showed results consistent with those of the total sample, including males. Healthy adults and children were not assessed for ethical reasons. Therefore, these results should not be extrapolated to cohorts with different characteristics. Furthermore, the original study was designed to determine changes, regarding pain levels and mouth opening limitation, in participants who underwent real treatment versus a placebo, but not directly to assess the asymmetry of bone structures. However, it provides access to information that is not accessible from healthy people. This information can only be obtained when there is an indication for CBCT, which is a per se invasive procedure, as was the case in the present study.

This study also has some methodological issues. The aim was to assess a clinical sign trichotomically (i.e., registering the mandible’s interincisive midline shift from the upper one, whether to the right or to the left, as a right, left, or no deviation), as it is a sign easily observable on dental inspection. Nevertheless, although the interincisive midline is deviated, it should not be interpreted or assumed a priori to represent retrognathia on the side towards which the mandibular incisors are deviated. This could also be explained as a possible displacement of the teeth, but not necessarily of the basal bone.

This study does not permit determining the cause/s of SS asymmetry. Further longitudinal research, with a larger sample size, could elucidate the cause as a means to treat/prevent asymmetries of the SS.

## 5. Conclusions

The side to which the jaw interincisive midline deviates did not predict less development of the hemimandible. Unexpectedly, this study shows evidence for the asymmetry of mandible length: the mandible was larger on the right side than on the left side; we speculate that this asymmetry could be attributed to the hemispheric dominance. Future studies may serve to identify the possible cause of this asymmetry.

## Figures and Tables

**Figure 1 diagnostics-15-00161-f001:**
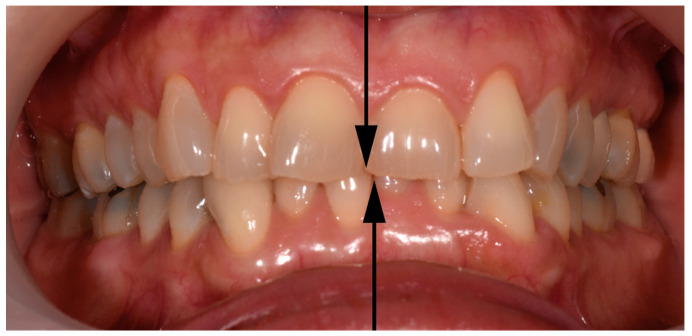
A typical jaw dental midline shift (toward the left side), where a larger right hemimandible can be expected.

**Figure 2 diagnostics-15-00161-f002:**
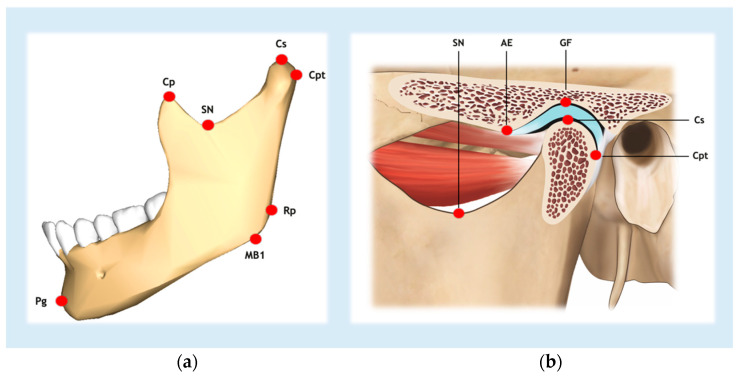
Hard tissue landmarks: (**a**) jaw landmarks; (**b**) temporomandibular joint with reference points.

**Figure 3 diagnostics-15-00161-f003:**
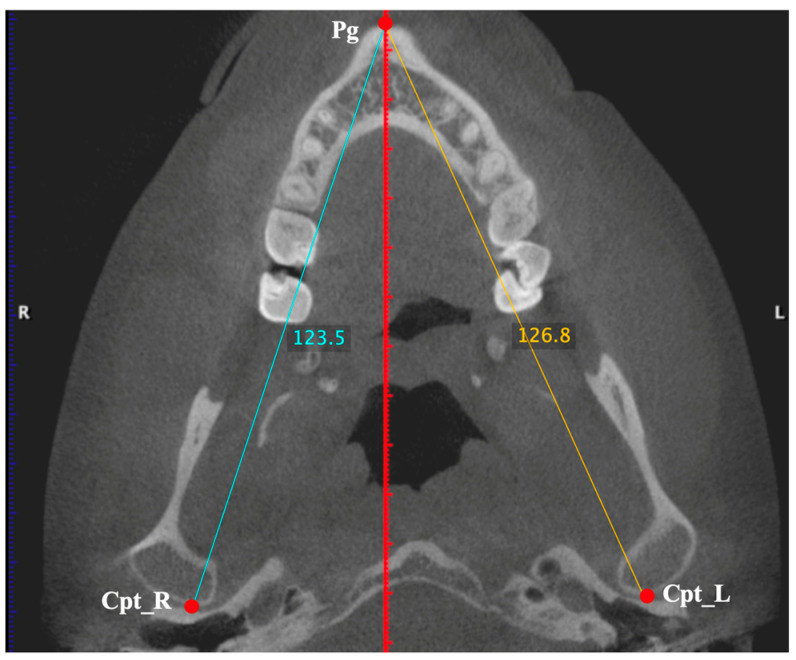
Measurements of the lengths of both hemimandibles (blue or yellow lines).

**Table 1 diagnostics-15-00161-t001:** Landmarks in the CBCT analysis.

**Abbreviation**	**Reference**	**Definition**
**Reference Points**		
AE	Articular eminence	Most inferior point of the apex of the articular eminence
ANS	Anterior nasal spine	Most anterior point of the nasal spine, at the margin of the piriform aperture
Cl	Condyle-lateral	Most lateral point of the mandibular condyle
Cp	Coronoid	Most superior point of the coronoid process
Cpt	Condyle-posterior	Most posterior point of the mandibular condyle
Cs	Condyle-superior	Most superior point of the mandibular condyle
GF	Glenoid fossa	Most superior point of the glenoid fossa
MB1	Mandibular ramus	Most inferior point of the inferior border of the mandibular ramus
Or	Orbital	Lowermost point of the orbital margin
Pg	Pogonion	Most anterior point of the outline of the mandibular symphysis
P16	Tooth 16	Most posterolateral point of the crown of tooth 16
P26	Tooth 26	Most posterolateral point of the crown of tooth 26
PNS	Posterior nasal spine	Most posterior point of the nasal spine, at the margin of the piriform aperture
Po	Porion	Most anterior point of the mandibular symphysis
Rh	Rhinion	Midline point at the inferior margin of the internasal suture
Rp	Mandibular angle	Most prominent posterosuperior point of the angle of the mandible at the mandibular ramus
SN	Sigmoid notch	Most inferior point of the sigmoid notch
**Reference planes**		
Abbreviation	Reference	Definition
FP	Frankfort horizontal plane	Plane passing through the porion point (Po) and the bilateral orbital (Or)
MSP	Mid-sagittal plane	Plane joining the anterior nasal spine (ANS), the posterior nasal spine (PNS), and the rhinion (Rh)

**Table 2 diagnostics-15-00161-t002:** Biological characteristics of the participants (*n* = 38). IQR, interquartile range; SD, standard deviation; VAS, 0–10 visual analog scale.

Characteristic	Participants
Female sex—no. (%)	35 (92.1)
Median age (IQR)—years	29.5 (25–42)
Affected side/s—no. (%)	
Right	7 (18.4)
Left	19 (50)
Both	12 (31.6)
Arthralgia (with or without myalgia)—no. (%)	32 (84.2)
Myalgia (without arthralgia)—no. (%)	6 (15.8)
Jaw-pain score (VAS)	
Median (IQR)	7 (6–7.6)
Mean (SD)	6.7 (1.2)
Habitual chewing side—no. (%)	
Right	12 (31.6)
Left	18 (47.4)
Alternate	8 (21.1)
Side toward lower dental midline shift was deviated—no. (%)	
Right	16 (42.1)
Left	13 (34.2)
Both	8 (21.1)
Cannot be assessed	1 (2.6)
Condylar path angles in relation to Frankfort Horizontal Plane	
Right side: Mean (SD) (degrees)	50.5 (10.7)
Left side: Mean (SD) (degrees)	50.1 (10.5)

**Table 3 diagnostics-15-00161-t003:** Dimensions of bone structures from CBCT images (*n* = 38) and intraindividual (right versus left side) comparisons, with the effect size values (Cohen’s d). A positive value of mean differences indicates larger dimensions on the right side and vice versa. SD—standard deviation; CI—confidence interval; *—two-tailed Student’s *t*-test for paired samples.

Variable Mean (SD)	Side		Paired Differences (95% CI)	*p*-Value *	Effect Size
	Right	Left			
Hemimandible length (mm)	119.4 (5.7)	118.6 (5.3)	0.9 (0.2 to 1.5)	0.01	0.4
Mandibular body length (mm)	92.2 (5.2)	92.9 (5.1)	−0.7 (−1.4 to 0.1)	0.08	−0.3
Mandibular ramus height (mm)	65.6 (5.4)	64.5 (4.3)	1.1 (0.1 to 2.2)	0.03	0.4
Coronoid process height (mm)	11.7 (3.2)	11.6 (3.4)	0.1 (−0.3 to 0.5)	0.51	0.1
Condylar process height (mm)	16.0 (2.3)	15.4 (2.1)	0.7 (0.0 to 1.3)	0.04	0.4
Distance Cl-MSP (mm)	56.5 (3.0)	56.5 (3.2)	0.0 (−0.8 to 0.8)	0.97	0.9
Hemimaxilla width (mm)	27.0 (1.3)	27.0 (1.6)	0.0 (−0.5 to 0.5)	0.99	0.0
AE height (mm)	7.0 (1.7)	6.6 (1.5)	0.5 (0.0 to 0.9)	0.04	0.4
AE inclination angle (degrees)	33.5 (7.1)	33.7 (6.5)	−0.2 (−2.0 to 1.6)	0.80	0.0
Condylar path angle (degrees)	45.7 (9.9)	48.8 (9.7)	−3.1 (−6.3 to 0.0)	0.05	−0.3

**Table 4 diagnostics-15-00161-t004:** Bone dimensions measured on CBCT images (*n* = 29). Intra-individual comparisons between the side towards which the lower dental midline is deviated (ipsilateral) and the opposite side (contralateral). Positive values of the differences mean that the dimensions are larger on the side to which the mandibular dental midline is deviated; negative values indicate that it is larger on the side opposite to that on which the lower dental midline is deviated. Results for Student’s *t*-test; SD; *p*-value and 95% coefficient interval (CI) and the effect size values (Cohen’s d) are presented.

Variable Mean (SD)	Side Towards the Jaw Incisal Midline Deviates		Paired Differences (95% CI)	*p*-Value	Effect Size
	Ipsilateral	Contralateral			
Hemimandible length (mm)	118.8 (5.1)	118.8 (5.4)	−0.2 (−1.1 to 0.6)	0.60	−0.1
Mandibular body length (mm)	92.1 (4.4)	92.5 (4.7)	−0.4 (−1.4 to 0.6)	0.41	−0.1
Mandibular ramus height (mm)	65.4 (5.1)	64.9 (5.2)	0.5 (−0.7 to 1.6)	0.44	0.1
Coronoid process height (mm)	11.7 (3.3)	12.1 (3.7)	−0.4 (−0.9 to 0.0)	0.05	−0.4
Condylar process height (mm)	15.7 (1.8)	16.1 (2.3)	−0.3 (−1.0 to 0.4)	0.40	−0.2
Distance Cl-MSP (mm)	56.9 (3.2)	56.5 (3.2)	0.4 (−0.5 to 1.3)	0.35	0.2
Hemimaxilla width (mm)	27.1 (1.3)	27.0 (1.3)	0.0 (−0.4 to 0.5)	0.87	0.03
AE height (mm)	7.0 (1.8)	6.8 (1.5)	0.2 (−0.4 to 0.8)	0.47	0.1
AE inclination angle (degrees)	34.3 (7.1)	33.7 (7.0)	0.6 (1.5 to 2.7)	0.56	0.1
Condylar path angle (degrees)	47.5 (10.8)	48.9 (9.1)	0.6 (−1.5 to 2.7)	0.57	0.1

**Table 5 diagnostics-15-00161-t005:** Bone dimensions from CBCT images (*n* = 25) on the sides affected versus unaffected (with chronic unilateral TMD pain). Comparisons were carried out by two-tailed Student’s *t*-test for paired samples. A positive value of differences means that measurements are higher on the affected side, and vice versa. D—standard deviation; CI—confidence interval.

Variable Mean (SD)	Chronic TMD (Unilateral Symptoms)		Paired Differences (95% CI)	*p*-Value	Effect Size
	Affected Side	Unaffected Side			
Hemimandible length (mm)	119.8 (5.6)	120.1 (6.3)	−0.3 (−1.2 to 0.6)	0.54	−0.1
Mandibular body length (mm)	93.3 (5.9)	93.2 (5.9)	0.0 (−1.0 to 0.6)	0.87	−0.1
Mandibular ramus height (mm)	65.2 (4.4)	66.5 (5.7)	−1.3 (−2.6 to 0.05)	0.06	−0.4
Coronoid process height (mm)	11.8 (3.9)	11.6 (3.7)	0.3 (−0.2 to 0.7)	0.25	0.2
Condylar process height (mm)	15.7 (2.2)	16.2 (2.7)	−0.5 (−1.3 to 0.4)	0.31	−0.2
Distance Cl-MSP (mm)	57.3 (3.2)	56.8 (1.7)	0.5 (−0.5 to 1.5)	0.35	0.2
Hemimaxilla width (mm)	27.1 (1.7)	27.1 (1.7)	0.0 (−0.6 to 0.5)	0.92	1.4
AE height (mm)	6.5 (1.6)	7.1 (1.9)	−0.5 (−1.2 to 0.1)	0.09	0.4
AE inclination angle (degrees)	32.6 (7.7)	33.3 (7.1)	−0.7 (−1.9 to 0.5)	0.25	−0.2
Condylar path angle (degrees)	48.9 (9.5)	45.2 (11.0)	3.7 (0.3 to 7.2)	0.04	0.4

## Data Availability

The datasets generated during and/or analyzed during the current study are available from the corresponding author on reasonable request. Correspondence and requests for materials should be addressed to U.S.P.

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
