# Peer review of "Clinical Dental Midline Shift Is Not a Predictor of the Side of Shorter Hemimandible: A Cone Beam Computed Tomography Diagnostic Study"

_diagnostics, 2025, doi:10.3390/diagnostics15020161_

Round 1

Reviewer 1 Report

Comments and Suggestions for Authors

Dear Authors,

congratulations on your interesting work, which offers potential significant perspectives for future analyses.

Nevertheless, I would like to highlight a few points of interest or offer some clarifications that I believe could enhance the quality of your article.

1)      It would be advisable to amend the title to specify that the term 'midline shift' is related to the dental midline.

2)      As well known, asymmetries can recognise different groups of causes. Asymmetries of a skeletal nature are mostly due to jaw growth problems, such as hemimandibular elongation (HE). Dental asymmetries may occur with perfect facial symmetry, e.g. the early loss of a tooth in the anterior region may cause the lateral migration of other teeth. Finally, there are functional asymmetries in which the mandible deviates laterally during the dental closure due to abnormal teeth contacts. Did you consider dental asymmetries as the sole parameter and then check whether there was also a skeletal problem (shorter midline on the affected side)? Please better explain these data in the text and include a general description on the classification of asymmetries.

3)      Please clarify why the dental midline of the upper jaw was selected as the reference point rather than face midline. Was the coincidence of the two midlines initially verified? or did you only want to consider dental midlines?

4)      Could you please explain how you determined the sample size and how you decided on the number of participants?

Please, also consider the further observations:

Lines 50-51: Could you provide a more detailed argument regarding the meaning of these lines, with particular reference to the concept of 'cerebral lateralisation'?

Lines 141-143: In the absence of a universally accepted reference system in 3D cephalometry, please specify which FH and MSP plans were used and provide the bibliographical reference.

Lines 145-147: Please clarify whether the anthropometric points were affixed to the 3D reconstructions or to the different cuts of the CBCT.

Line 229: It is likely that the term 'left side' has been included twice in the text.

Line 336: Please provide the rationale behind the assumption that the deviation of the median is related to the predominance of one hemisphere. Further clarification would be appreciated.

Author Response

Dear Reviewer,

Thank you very much for approaching this review with a critical and constructive purpose.

Changes in the text are highlighted in red.

We have attempted to respond to each of the questions. Specifically:

  • It would be advisable to amend the title to specify that the term 'midline shift' is related to the dental midline.

Yes, certainly. Thank you very much. The word “dental” was added in the title and throughout the text.

  • As well known, asymmetries can recognise different groups of causes. Asymmetries of a skeletal nature are mostly due to jaw growth problems, such as hemimandibular elongation (HE). Dental asymmetries may occur with perfect facial symmetry, e.g. the early loss of a tooth in the anterior region may cause the lateral migration of other teeth. Finally, there are functional asymmetries in which the mandible deviates laterally during the dental closure due to abnormal teeth contacts. Did you consider dental asymmetries as the sole parameter and then check whether there was also a skeletal problem (shorter midline on the affected side)? Please better explain these data in the text and include a general description on the classification of asymmetries.

Yes, we considered dental asymmetries as the main parameter and subsequently assessed whether there was also a skeletal problem. Please note that the main complaint was orofacial pain. We also speculated that the hemimandibles on the affected side are shorter than those on the unaffected side. We appreciate the reviewer’s clarification.

The following paragraphs were included:

Lines 41-46: “Facial asymmetry may have different origins, namely: i) congenital (originating prenatally), ii) developmental (originating during growth), and iii) acquired (resulting from functional jaw displacements, traumatic injury, or other factors) (Srivastava et al., 2018). However, dental asymmetries may occur with perfect facial symmetry. According to Bishara et al. (1994), facial asymmetry can be classified as dental, skeletal, muscular, functional, or combinations thereof.”

Lines 78-81: “In this study, we considered clinical dental asymmetries as the sole parameter and then evaluated whether there was also a skeletal problem. We also speculated that the hemimandible on the affected side was shorter than that on the opposite (unaffected) side.”

Lines 123 to 129: “All participants in the study were carefully evaluated both clinically and through computerized kinesiography (K7I, Myotronics Inc.) from a biodynamic perspective. The displacement between centric occlusion and maximum intercuspation occurred at the expense of mandibular advancement, but with minimal or no displacement in the latero-medial direction. Therefore, the asymmetry of the lower dental midline relative to the upper midline was organic in this study population (Bishara et al., 1994).”

  1. Please clarify why the dental midline of the upper jaw was selected as the reference point rather than the facial midline. Was the coincidence of the two midlines initially verified, or did you only consider dental midlines?

The reviewer is correct in noting that the facial midline is particularly important, especially when assessing asymmetry and its influence on aesthetics and treatment planning for surgery or extensive rehabilitations. This study included individuals with TMDs but excluded those with complaints specifically related to asymmetry or aesthetics, where the differences tend to be more pronounced.

The assessment of asymmetry relative to the sagittal plane requires CBCT, which is not routinely indicated in clinical settings unless required for treatment planning due to its potential invasiveness. The evaluation of the dental midline and the deviation of the lower midline relative to the upper midline can be performed clinically. As the reviewer correctly pointed out, verifying the relationship between the upper and lower dental midlines during centric relation and maximum intercuspation is essential for distinguishing organic from functional deviations and for detecting premature occlusal contacts responsible for mandibular deflective closure. This aspect was carefully evaluated during the clinical trial, in which a criterion for inclusion was normal or near-normal occlusion (Angle Class I, stable). This study confirms the reviewer’s statement (point 2): “Dental asymmetries may occur with perfect facial symmetry,” which has been included in the manuscript (please see line 42).

Yes, there were reasons for selecting the dental midline of the upper jaw as the reference point rather than the facial midline. The upper jaw remains static during mandibular movements. Mandibular movements stimulate growth centers and influence the growth of each hemimandible. Functional factors, therefore, have a more direct impact on mandibular growth than on the maxilla. This may explain why most facial asymmetries are observed in the lower third of the face (Severt & Proffit, 1997). Additionally, a primary objective of this study was to determine whether mandibular asymmetry (defined as a lack of mirror image between the right and left sides of the mandible) was related to dental asymmetry or TMDs.

Added sentence:

Lines 266-269: “Most asymmetries occur in the lower face (Severt and Proffit, 1997); therefore, the dental midline of the jaw was selected as a reference point rather than the facial midline to explore whether different interarch incisal asymmetries can be observed in a clinical context, although not primarily related to facial aesthetics.”

  1. Could you please explain how you determined the sample size and how you decided on the number of participants?

The sample size was determined based on convenience. We aimed to avoid harming participants. CBCT is an ionizing procedure and should only be performed to improve diagnosis and establish an accurate therapy plan. All available eligible CBCTs were included. This approach precluded the inclusion of a control group of healthy individuals. Similar studies have used comparable sample sizes (e.g., Miresmaeili et al., 2021; Yanez-Rico et al., 2013; Tun Oo et al., 2022).

The words “The sample…was chosen by convenience…” were added to the text (Line 215).

Please also consider the following observations:

Lines 50-51: Could you provide a more detailed explanation of these lines, particularly concerning the concept of ‘cerebral lateralization’?

Because the descending central motor fibers cross to the opposite side at the bulbar level, the preference for handedness is usually toward the side contralateral to the dominant hemisphere. The main fibers of the masticatory nuclei of the fifth cranial nerve, although there are interconnections between both sides of the nuclei, are directed to the same side of the masticatory system. According to this concept, masticatory function may be more efficient on the dominant hemispheric side. This hypothesis, although plausible, is speculative.

Lines 141-143: In the absence of a universally accepted reference system in 3D cephalometry, please specify which FH and MSP planes were used and provide the bibliographical reference.

The definitions of the main reference planes, according to Tun Oo et al., (2022) has been included in the Table 1.

Lines 145-147: Please clarify whether the anthropometric points were affixed to the 3D reconstructions or the individual CBCT slices.

The anthropometric points were identified in the 3D reconstructions and, subsequently, they were transferred to the different slices obtained in each CBCT.

Line 229: It is likely that the term ‘left side’ has been included twice in the text.

Yes, you are correct. The second instance of “left” was replaced with “right” in the text (Line 232). Thank you.

4.2 Line 336: Please provide the rationale for the assumption that the median deviation is related to hemispheric dominance. Further clarification would be appreciated.

Hemispheric dominance contributes to the preferential use of the side of the body (e.g., legs or arms) opposite the dominant hemisphere. The spinal tract crosses to the opposite side at the medulla, while neurons from the trigeminal nuclei project ipsilaterally to the viscerocranium via the trigeminal nerve. This could result in the preferential use of the dominant side for chewing.

Reviewer 2 Report

Comments and Suggestions for Authors

Clinical midline shift is not a predictor of the side of shorter hemimandible. A CBCT diagnostic study. 

In the abstract: The first two lines in the results section, better to be sent to the methodology (abstract section). Still in the abstract section: what do you mean by “Mean age 34.6(11.9) years old”?

The aim of the study is clear.

The title of the article is appropriately selected but did not denote the study performed.

In all the manuscript the midline mentioned is confusing: mandibular or maxillary.

The references are recent (2024) and relevant in addition to this are well written and arranged and highly related to the study.

The authors reviewed well what is already written and investigated in the subject selected.

The methodology is clear enough and the total number is fair: but I am wondering why 35 females and only 3 males.

Major point in Methodology: Either to restrict the study for FEMALES only or to increase the number of males participants.

Thanks

Author Response

Dear Reviewer,
Thank you very much for the critical and constructive evaluation.
We hope we have been able to respond to your comments
Results Section in Abstract
The first two lines in the results section, better to be sent to the methodology (abstract section)
Done. Thank you.
Still in the abstract section: what do you mean by “Mean age 34.6(11.9) years old”?
"Mean age 34.6(11.9) years old" was replaced with "Mean (SD) age 34.6 (11.9) years." Thank you for pointing this out.
The aim of the study is clear.
Thank you.
The title of the article is appropriately selected but did not denote the study performed.
The word “dental” was added to the title. Thank you.
In all the manuscript the midline mentioned is confusing: mandibular or maxillary.
This aspect was clarified throughout the text. Thank you.
The references are recent (2024) and relevant in addition to this are well written and arranged and highly related to the study.
Thank you.
The authors reviewed well what is already written and investigated in the subject selected.
Thank you.
The methodology is clear enough and the total number is fair: but I am wondering why 35 females and only 3 males.
Major point in Methodology: Either to restrict the study for FEMALES only or to increase the number of males participants.
The Reviewer’s concern is entirely valid. Bone size is generally larger in males than in females, which justifies assessing sex differences. However, excluding males entirely could mask the reality that females are usually, though not exclusively, the ones seeking therapy. This highly asymmetrical female-to-male ratio is characteristic of samples from patients with chronic TMDs referred to tertiary care centers (Stohler and Zarb, J. Orofac. Pain 1999), as observed in our study.
A post-hoc test was performed as a sensitivity analysis, including only females (N=35), and it yielded similar results (Table 3’) to the total sample including males (Table 3). While bone size differences between sexes are evident, the study focuses on side differences or asymmetry (intra-individual comparisons), rather than absolute measurements of size.
Discussion Section
The following paragraph was added to the discussion section (Lines 322–330):
"This study could be limited to female participants due to the low number of males. However, this asymmetrical female-to-male ratio is characteristic of samples from patients with TMDs referred to tertiary care centers (Stohler and Zarb, 1999). Comparisons of anatomical structure sizes must often account for sex differences. However, in the study of asymmetry (intra-individual comparisons), no a priori differences between sexes are expected. The main purpose of this study is to answer the clinical question: Is the clinical perception that the mandible is smaller on the side to which the mandibular midline deviates accurate? A post-hoc test, including only females (N=35), showed results consistent with those of the total sample.”
Table 3’. Dimensions of bone structures from CBCT images (n=35) and intraindividual (right versus left sides) comparisons. Additionally, the effect size values (Cohen’s d) are presented. A positive value of mean differences indicates larger dimensions on the right side and vice versa. SD – Standard deviation; CI – Confidence Interval; * – Two-tailed Student’s t test for paired samples.

Variable
Mean (SD)    Side    Paired differences
(95% CI)    P-value *    Effect size
    Right    Left            
Hemimandible length (mm)    118.8 (4.4)    118.0 (4.3)    0.8 (0.1 to 1.5)    0.03    0.4
Mandibular body length (mm)    91.3 (3.5)    92.1 (3.9)    -0.8 (-1.6 to 0.0)    0.04    -0.3
Mandibular ramus height (mm)    65.1 (4.9)    64.1 (4.2)    1.0 (0.0 to 2.0)    0.05    0.3
Coronoid process height (mm)    11.4 (2.8)    11.3 (3.2)    0.0 (-0.3 to 0.5)    0.75    0.1
Condylar process height (mm)    16.2 (2.2)    15.6 (2.1)    0.7 (0.0 to 1.3)    0.05    0.3
Distance Cl-MSP (mm)    56.4 (2.7)    56.4 (3.0)    0.0 (-0.8 to 0.8)    0.99    -0.0
Hemimaxilla width (mm)    26.8 (1.1)    26.8 (1.4)    0.0 (-0.4 to 0.5)    0.84    0.0
AE height (mm)    7.1 (1.7)    6.6 (1.5)    0.5 (0.0 to 0.9)    0.05    0.3
AE inclination angle (degrees)    34.0 (7.2)    33.9 (6.6)    -0.1 (-2.0 to 2.0)    0.93    0.0
Condylar path angle (degrees)    45.6 (10.1)    48.6 (10.2)    -3.0 (-6.4 to 0.4)    0.08    -0.3

The reference to Stohler and Zarb, J. Orofac. Pain 1999 was included in the discussion-study limitations and also in the References sections.
We respectfully ask the Reviewer to consider maintaining the total sample size. Raw data has been included in the supplementary appendix, allowing readers to evaluate potential confounding factors in detail.

Round 2

Reviewer 1 Report

Comments and Suggestions for Authors

I would like to thank the authors for the corrections made which, in my opinion, are sufficiently detailed

Reviewer 2 Report

Comments and Suggestions for Authors

Thanks for addressing my points